# Extracellular Vesicles in Multiple Sclerosis: Their Significance in the Development and Possible Applications as Therapeutic Agents and Biomarkers

**DOI:** 10.3390/genes15060772

**Published:** 2024-06-12

**Authors:** Ida Manna, Selene De Benedittis, Danilo Porro

**Affiliations:** 1Institute of Molecular Bioimaging and Physiology (IBFM), National Research Council (CNR), Section of Catanzaro, 88100 Catanzaro, Italy; 2Institute for Biomedical Research and Innovation (IRIB), National Research Council (CNR), 87050 Cosenza, Italy; 3Institute of Molecular Bioimaging and Physiology (IBFM), National Research Council (CNR), Segrate, 20054 Milan, Italy

**Keywords:** multiple sclerosis, inflammation, extracellular vesicles, biomarkers, drug delivery

## Abstract

Extracellular vesicles (EVs) are “micro-shuttles” that play a role as mediators of intercellular communication. Cells release EVs into the extracellular environment in both physiological and pathological conditions and are involved in intercellular communication, due to their ability to transfer proteins, lipids, and nucleic acids, and in the modulation of the immune system and neuroinflammation. Because EVs can penetrate the blood–brain barrier and move from the central nervous system to the peripheral circulation, and vice versa, recent studies have shown a substantial role for EVs in several neurological diseases, including multiple sclerosis (MS). MS is a demyelinating disease where the main event is caused by T and B cells triggering an autoimmune reaction against myelin constituents. Recent research has elucidate the potential involvement of extracellular vesicles (EVs) in the pathophysiology of MS, although, to date, their potential role both as agents and therapeutic targets in MS is not fully defined. We present in this review a summary and comprehensive examination of EVs’ involvement in the pathophysiology of multiple sclerosis, exploring their potential applications as biomarkers and indicators of therapy response.

## 1. Introduction

Extracellular vesicles (EVs) are small particles enveloped in a lipid bilayer secreted from almost all cells, and they can be found in different types of biofluid, including serum, plasma, urine, saliva, and cerebrospinal fluid (CSF). EVs transport lipids, proteins, and genetic material to nearby and distant cells as parts, and represent an alternative mechanism for cell-to-cell communication [1]. According to their size and origin, they are divided into three primary subgroups: exosomes (50–100 nm); microvesicles (100–1000 nm); and apoptotic bodies (100–5000 nm). Exosomes are the smallest vesicles that are released from inside the cell and follow the endosomal pathway; the microvesicles are also released from inside the cell but are produced using the cell membrane itself, which incorporates the load to be transported and “buds” towards the outside; and apoptotic bodies are a sign of dying cells. The latter have received less attention in the field of applications, whereas exosomes and microvesicles share a common characteristic of transporting RNA, proteins, and lipids, indicating their role in the regulation of diverse biological functions through distinct molecular mechanisms. Cell culture media and biological fluids can be used to isolate EVs, and to identify and separate EVs, various techniques are used, mostly based on the proteins and lipids that are expressed [2]. Table 1 summarizes the most common techniques for separating EVs, together with both their advantages and disadvantages.

The understanding of the regulatory characteristics of the immune and neurological systems has been completely transformed by the identification of EVs as a novel form of intercellular communication. EVs play a significant part in immune control and are secreted by both immune and nonimmune cells

Research has demonstrated, for instance, that EVs produced from endothelium cells, B lymphocytes, and dendritic cells stimulate and activate T cells through the presentation of antigens, thus serving as a promoter of the adaptive immune response. EVs have been linked to the pathophysiology of inflammatory, autoimmune diseases due to their effect on the immune system [3,4]. Interestingly, activated microglia, due to high extracellular concentrations of ATP, a chemical released by damaged cells, release EVs [5,6]. Given that neurons, glia, and peripheral immune cells form an integrative network to actively regulate immunological processes that affect brain functions, it is not surprising that EVs contribute in the etiology of many illnesses [7]. Among them, multiple sclerosis (MS) is a classic disease that severely impairs immune system and central nervous system (CSN) connection [8,9]. A number of researchers have begun to consider extracellular vesicles as potential diagnostic and prognostic biomarkers of MS, as EVs are emerging as significant mediators of both pathological and reparative mechanisms in the neurological disease. Their bidirectional trafficking from the CNS in extra-CNS biological fluids is facilitated by blood–brain barrier (BBB) leaks associated with MS pathophysiology [10,11]. In this review, we provide an overview and a detailed discussion of the role of EVs in the pathogenesis of MS, as well as look at how they might be used as biomarkers and potential indicators for therapeutic response.

## 2. Multiple Sclerosis

MS is a neurodegenerative and neuroinflammatory disease of the CNS, where inflammation, demyelination, and axonal degeneration are, indeed, the main pathological signs. Although the exact cause of MS remains unknown, it is clear that an autoimmune response against the CNS is brought on by a complicated interaction of genetic, environmental, and epigenetic variables [12,13]. Progressive BBB breakdown that results in peripheral pathogenic T and B cells, antibodies, monocytes, and inflammatory mediators infiltrating the CNS is an essential aspect of the pathogenesis of MS. Leucocyte infiltration sets off a series of inflammatory events that cause demyelination, axonal damage, synaptic loss, and dysfunction, a condition known as synaptopathy, which culminates in neurodegeneration [14,15].

MS presents with a wide range of clinical manifestations, from slightly debilitating to severe forms that result in progressive, irreversible impairments in cognition and clinical functioning along with a limited treatment response [16]. Patients with relapsing remitting (RR) MS might have neurological dysfunctional episodes with or without permanent impairment. RRMS is the most prevalent clinical type of the disease among the different patterns. It is characterized by multiple episodes of relapse, brought on by autoimmune aggressiveness, and periods of remission, brought on by the shutdown of the immune system [17,18]. Among those with RRMS, 15–30% will develop to secondary progressive MS (SPMS), which causes progressive disability. In conclusion, around 15% of people have primary progressive (PP) MS, which deteriorates neurologic function from the moment symptoms arise without relapses or remissions [19]. Many clinical features, including the clinical examination, magnetic resonance imaging, analysis of CSF, and electrophysiology, are important in the diagnosis and monitoring of MS. To diagnose and monitor the progression of multiple sclerosis, there are currently no valid biomarkers.

## 3. Dual Role of EVs in MS: Neurodegenerative and Neuroprotective

In MS, due to the disruption of immune system and central nervous system connection, EVs may be important in both pathogenic and reparative pathways. There is evidence that the majority of cells, including mast cells, oligodendrocytes, glial cells, neural cells, and astrocytes, release EVs, indicating their active role in the CNS [20]. As mentioned previously, MS is the first neurological condition for which EVs have been identified. First, Scolding et al. reported the presence of EVs in MS. Additionally, they demonstrated that during oligodendrocyte recovery from injury, EVs were liberated from cell surfaces [21]. The functional effects and therapeutic potential of EVs in MS is a rapidly evolving topic [22,23]. All types of CNS cells secrete EVs that promote neuronal trophic support, synaptic plasticity, and myelination under normal conditions [24]. EVs influence the etiology, progression, and/or recovery of MS by mediating neuroinflammatory responses, controlling tissue damage, and repair in response to CNS injury [25]. Via a number of pathways, including translocation over the damaged and undamaged BBB, the blood–cerebrospinal barrier located in the brain ventricles, and the passage through CSF, circulating EVs can move from the blood flow to the central nervous system tissue and in the opposite direction [26]. Accordingly, EVs can be crucial in the reciprocal communication between the CNS and the peripheral, leading to a complicated cellular crosstalk mediated by EVs.

Conversely, in the pathophysiology of MS, EVs may have a neuroprotective function. According to specific hypotheses, EVs play a critical role for preserving myelination, repairing damaged neurons, and controlling synaptic plasticity in the CNS.

EVs may specifically exhibit beneficial effects on synaptic activation [27,28]. In Antonucci et al.’s work, findings revealed the participation of neuronal sphingosine in microglia-to-neuron signaling pathway and identified microglia-derived EVs as a novel mechanism by which microglia influence synaptic activity [29]. Bhargava et al. evaluated pre- and post-synaptic proteins in neuronal-enriched extracellular vesicles (NEVs) and complement components in astrocytic-enriched extracellular vesicles (AEVs), to ascertain if NEVs and AEVs provide biomarkers indicating complement-mediated synaptic loss in MS in comparison to controls. They discovered that AEVs from MS patients had higher amounts of multiple complement cascade and lower levels of NEV synaptophysin and synaptopodin. These results demonstrate the potential use of circulating EVs to detect synaptic loss in multiple sclerosis and imply a connection between astrocytic complement production and synaptic loss [30]. Since EVs carry functional mitochondrial components, Ladakis et al. investigated the activity of mitochondrial components in NEVs from MS patients based on imaging results and assessed the potential of these measurements to predict disease progression. The findings of this pilot investigation provide support for larger follow-up longitudinal studies by indicating that mitochondrial measurements in circulating NEVs may function as possible biomarkers of illness progression [31].

The presence of proteins linked to the creation of synaptic buttons in the cargo of EVs provides additional evidence about their role in promoting synaptic plasticity [32]. It has been proposed that specific EV-derived RNAs, in addition to proteins, may fulfill a number of roles in the processes involved in plasticity. Neuron-derived EVs have recently been revealed to contain mRNA associated with the activity-regulated cytoskeleton-associated protein, a master synaptic plasticity regulator [33].

It has been shown that EVs play a role in the process of myelin biogenesis as well. In fact, it has been demonstrated that myelin proteins and particular RNA for the development of myelination processes are present in oligodendrocyte-derived EVs [34]. Furthermore, it has been shown that synapsin 1 is abundant in astrocyte-derived EVs, which may help nerve cells proliferate, survive, and differentiate throughout development [35]. When considered collectively, the data point to the active involvement of EVs in myelination, oligodendrocyte proliferation, and neuronal development.

In any case, as CNS-derived EVs penetrate peripheral circulation, they serve as useful therapeutic targets and diagnostic tools in addition to being a readily available biomarker source for remotely analyzing the state of the CNS [36].

To sum up, while our understanding of EVs’ involvement in MS is still restricted, their potential relevance in the disease’s pathophysiology presents new opportunities for the assessment of disease biomarkers, treatment selection, and efficacy.

## 4. EVs in MS Pathogenesis

According to a number of studies that this review summarizes, EVs have a direct role in the pathophysiological progression of this disease.

The sections that follow will cover these topics. The characteristics of the reported EVs are listed in Table 2.

Damage to the BBB therefore represents the earliest event of MS and is closely associated with the upregulation of adhesion molecules by cytokines produced by T lymphocytes and macrophages. Subsequently, this phenomenon could favor a second wave of influx of largely non-antigen-specific T lymphocytes at the lesion level [43]. Although there is increasing evidence that EVs can traverse the BBB in both healthy and pathological settings, the biological processes that underlies this process are still not fully understood [44]. BBB opening in MS is typically linked to new lesions and, as a result, immune infiltration that starts in the vicinity of parenchymal microvessels. This infiltration is caused by a decrease in the expression or modification of proteins junctions, which increases leukocyte transmigration through the BBB, a process that is controlled by CAMs (ICAM-1, VCAM-1) and chemokine signaling processes [45]. EVs produced by various cell types in the peripheral and CNS have been reported to interact with endothelial cells and leukocytes to alter BBB function and induce immune system cell migration in neuroinflammatory conditions. Similar to this, stimulated endothelial cells generate a significant amount of EVs that can directly affect the BBB and increase its leakage. Endothelial EVs carry compounds that induce endothelial activation [37], metalloproteases that may facilitate BBB rupture [46], and compounds that support monocyte and lymphocyte transendothelial migration through the compromised BBB [38,39]. In Marcos-Ramiro et al.’s work, they provide a thorough examination of the circulating endothelium- and platelet-derived EVs in the plasma of the various clinical manifestations of MS. They discovered a large and comparable increase in all subtypes, including patients with typical clinically isolated syndrome (CIS) who had already recovered or those who were in the remission stage of the illness, as compared to normal control individuals. In particular, platelet-derived EVs from RRMS patients, compared to EVs from healthy donors, cause a greater disruption of the endothelium barrier. It is interesting that they discovered experimental evidence that suggests that plasma EVs cause malfunction of the human endothelium barrier and may therefore actively contribute to the course of MS [40]. Moreover, it has been shown that EVs generated by microvascular endothelial cells in the human brain have the ability to control immunological responses. In fact, in vitro tests supporting the activation of CD4+ and CD8+T cells revealed that endothelium-derived extracellular vesicles may deliver antigens when separated from stimulated human brain microvascular endothelial cells [39]. Stimulated endothelial cells were seen to release EVs that included metalloproteinases and caspase 1 when stimulated with proinflammatory cytokines such TNF-α, IFN-α, and IL-1 α. These cytokines and enzymes are well known for their capacity to cause the BBB to break down and to make it easier for myeloid and lymphocyte transmigration into the central nervous system [47,48]. Furthermore, activated microglia EVs store and release interleukin (IL)-1β [49], and MHC-II, propagating neuroinflammation [50]. It has been studied if circulating endothelial EV-monocyte conjugates occur in MS patients and whether they may play a role in the trans-endothelial migration of inflammatory cells. This is because monocytes and T cells are the main cell types present in the perivenular infiltrates that are typical of MS. The study demonstrated that endothelial EVs, specifically those carrying ICAM1, preferentially strongly bound to monocytes to form endothelial EV-monocyte conjugates. This was achieved by utilizing endothelial markers, CD54 and CD62E, in conjunction with a specific blood monocyte marker, CD11b, expressed on EVs during exacerbations. Moreover, endothelial EVs, which were produced by injecting TNF-α to brain microvascular endothelial cells, demonstrated the ability to stimulate monocytes and facilitate their trans-endothelial migration via a monolayer BBB model. Based on these data, it is possible that monocyte trans-endothelium migration involves EVs generated by activated endothelial cells [38]. Endothelial cells are known to release EVs when they activate or undergo apoptosis. Minagar et al., 2001 [37] used flow cytometry to determine the amount of endothelial microparticles (EMP) released into the plasma of MS patients and healthy controls. Furthermore, they examined the possibility that EMP could be lost by culturing microvascular endothelial cells (MVECs) from plasma of MS patients.

To investigate EVs as indicators of endothelial damage in MS, they used fluorescently labeled antibodies against two distinct endothelium antigens, CD31 and CD51, and they examined the plasma of 50 MS patients (30 in exacerbation and 20 in remission) and 48 controls for the in vitro investigations. When compared to healthy controls, the plasma of patients in exacerbation showed a 2.85-fold increase in CD31, which decreased to nearly normal levels during remission. On the other hand, in both flare and remission, CD51 stayed increased. This implies that whereas CD51 represents endothelial injury over time, CD31 is a sign of acute injury.

Additionally, they demonstrated that brain microvascular endothelial cells generated with plasma from MS patients in exacerbation released endothelial cells that carried the ICAM1 protein in comparison to patients who had clinically stable MS and healthy controls. This result agrees with endothelium activation [37]. In an effort to highlight the connection between MS patient plasma in exacerbation and monocyte trans-endothelial migration, it has been demonstrated that endothelial EV–monocyte conjugates markedly elevated the proportion of transmigrated cells in a BBB model [41].

Mazzucco et al. postulated that MS patients with active disease would have higher circulating amount of CNS endothelial-derived EVs. To investigate this, they designed a unique flow-cytometry-based technique for identifying EVs derived from CNS endothelial cells from MS patients. They discovered for the first time a technique to detect EVs derived from the endothelium of the central nervous system in human blood samples. The results of their pilot study suggest that elevated levels of CNS endothelial-derived EVs could be a biomarker for both active MS disease and BBB permeability [42].

A pathologic process known as the endothelial to mesenchymal transition, in which endothelial cells lose their specific role and de-differentiate into mesenchymal cells, has recently been detected on postmortem brain tissue from MS patients [51]. This mechanism, which has been linked to endothelial dysfunction, disrupts barrier function and vascular stability while upregulating mesenchymal markers. [52]. Presently, there is growing evidence that EVs, which are secreted by a particular subtype of glioblastoma cancer cells, play a role in the transition from endothelium to mesenchymal tissue [53]. Similar to MS, it is possible that EVs contribute to this process, although further research is needed to confirm this theory.

Although the brain is an immune-privileged area, it is possible that EVs formed from brain cells could distribute myelin antigens to the periphery, activating T cells before they enter the CNS. This is because EVs are able to penetrate the BBB [54]. Goetzl et al., 2019, found that individuals with Alzheimer’s disease had plasma astrocyte-derived EVs in their blood that were enriched with neuronal antigens, which is consistent with this theory [55]. In conclusion, EVs may contribute to the pathogenetic function of MS through a variety of pathways, including promoting T cell migration across the BBB, activating T cells during relapses, and increasing inflammation inside the CNS. On the other hand, there is still debate regarding the movement of autoantigens in the periphery.

## 5. EVs as Biomarkers for MS

The main components of EVs cargo include proteins, lipids, DNA, RNA, and noncoding RNA, whose sophisticated sorting mechanism reflects the state of the donor cell [56]. The most characteristic components of EVs are lipids, which include cholesterol, phospatidilserine, and sphingomielin, which constitute the EV membrane. Few studies have highlighted the presence of some lipid species in EVs, and EVs’ lipid composition seems to be regulated in relation to their parent cell, which shows a decrease in a number of lipid classes but not others [57]. Additionally, certain proteins that are inside and on the membrane of EVs are characteristic of the original cell [58]. EVs are suitable for the identification of CNS disease biomarkers due to their particular quality and their capacity to traverse the BBB. When it was discovered that EVs’ cargo included genetic material, particularly mRNAs and microRNAs, scientific interest in EVs increased. The discovery that the mRNA transcripts found in EVs differ significantly from those of the parent cell indicates that the packaging of EV cargo is selective and may even constitute a complex mechanism for genetic transfer across cells [59]. Similarly, several miRNAs that are of interest due to their control over gene expression appear to be selected only in EVs as they are not present in parental cells [60]. Surprisingly, miRNAs are the highest RNA component in EV cargo, indicating their important part in recipient cell biology [61]. miRNA’s delivery by EVs significantly increases their biological relevance because they have the ability to change the target cell’s genetic activity. Since miRNAs are highly expressed in cells of the immune system and CNS, it is plausible to think that they play a role in the pathogenetic pathways underlying disease. It is now understood that cargo, in any composition, can impact the biology of the host cell. In recent years, circulating EVs obtained popularity as potential biomarkers in liquid biopsy [62,63]. In addition to monitoring and predicting the clinical response to therapies, they can also be used as a diagnostic tool to find patients with early-stage diseases, assess the progression of the illness, and suggest a prognosis [64]. Regarding CSF research, Scolding et al. provided the first indication that EVs were present in MS patients’ CSF [21].

The following subsections will provide examples of current findings regarding the relationship between MS and EVs based on their cargo molecular.

### 5.1. EV-Derived Noncoding RNAs

Furthermore, a growing body of research suggests that EVs might carry and transmit small noncoding RNAs, in particular miRNAs, which are dysregulated in MS patients’ immune systems and central nervous systems and are being recognized as disease biomarkers. Indeed, research on EVs in MS patients has shown that, in addition to differences in relapse and treatment response, there is a general change in EV quantity and cargo as compared to controls [65,66,67]. Regarding this, EVs and their cargo have drawn interest recently as possible MS biological markers. The assessment of EV cargo in MS as a biomarker has been the topic of numerous investigations recently (Table 3).

Numerous RNA cargo components have been investigated in several disorders, which includes MS, and their significance has been shown for diagnosis [68], prognosis [83], and therapy response [69]. It is noteworthy that the majority of research has emphasized the importance of miRNAs. It has recently been determined that miRNAs, which are small, single-stranded, noncoding RNAs with 21–23 nucleotides, are post-transcriptional modulators of gene expression. They achieve this by either targeting the degradation of mRNA or blocking the translation of proteins. It has been demonstrated that a single miRNA can regulate the expression of numerous target mRNAs and that multiple miRNAs can regulate a single mRNA. Numerous biological functions, including cell division, proliferation, metabolism, apoptosis, inflammation, and immunology, depend on miRNAs. Neurodegeneration and autoimmunity are two significant human disorders that may be impacted by their manifestation [84]. With remarkable stability, miRNAs can be transported to distant target tissues by being secreted into the environment outside of cells [85]. These miRNAs may be present in EVs that function as messengers and facilitate communication between cells [86]. Since miRNA can alter the genetic behavior of the target cell, their transport by EVs greatly improves their biological relevance. Given that miRNAs are highly expressed in immunological and CNS cells, it is possible that they are involved in pathogenetic mechanisms that mediate the disease. In the last few years, a lot of effort has gone to understanding the role that miRNA plays in the pathogenesis of MS.

Two distinct studies that used next-generation sequencing (NGS) analysis of serum-derived EV microRNAs from MS patients have explained their association with disease state. Nine miRNAs were found to be differentially expressed in RRMS cases when compared to PPMS cases vs. healthy controls (HCs) [68]. On the other hand, Selmaj et al. demonstrated that four serum-derived EV miRNA were significantly reduced during relapse and radiological flare of RRMS patients [66]. In another study, EV miR-326, known to play a pathogenic function in MS, was shown to be significantly higher in RRMS patients compared to HCs [70]. Significantly, peripheral blood mononuclear cells (PBMCs) from relapsed patients expressed more miR-326 than those in remission and healthy subjects. Accordingly, miR-326 has been suggested as a biological marker of MS severity as it correlates with disease severity [87]. In the work of Manna et al., to determine if an EV-derived miRNA profile varies with IFN treatment in naïve, sensitive, and resistant RRMS patients, EV-associated miRNA profiling was carried out. Examination of miRNAs isolated from serum revealed sixteen miRNAs that were dysregulated in treated patients compared to nontreated patients: 14 were downregulated and 2 were upregulated. Notably, let-7miRNA, miR-451, miR-26a, miR-23a, miR-15b, miR-223, and miR-146a; however, some were already differentially implicated in MS [69,88]. Notably, the authors proposed that EV-derived miRNAs may be utilized to track the effectiveness of INF-therapy because miRNA deregulation was only verified in treated patients and, moreover, in responders. In the in vitro studies of Prada et al., they showed that EVs generated from inflamed microglia transfer the miR-146a-5p cargo into neurons. miR-146a-5p affects the expression of presynaptic and postsynaptic proteins, which are involved in the impact of dendritic spine development on synaptic integrity. It is restricted to microglia and is absent from hippocampus neurons [72]. Kimura and colleagues examined the overexpression of four miRNAs in microarray analysis of plasma-derived EVs from patients with HC and MS compared to healthy subjects. It is interesting to note that let-7i, which is overexpressed in MS patients, suppresses the induction of Treg cells by blocking the IGF1R/TGFBR1 pathway. Additionally, the frequency of Treg cells was lower in the group that had higher levels of EV-derived let-7i, which may have contributed to the pathogenesis of MS [71]. The role of erythrocyte-derived EV miRNAs in MS was examined in a relatively recent study, on purified erythrocyte-derived EVs. In the work of Groen et al., it was shown that erythrocyte-derived EVs are preferentially packaged and include the majority of miRNAs that are substantially expressed in red blood cells. Specifically, plasma from MS patients was found to have higher levels of miR-451a, which is transported by EVs from erythrocytes to endothelial cells. Indeed, due to their reduced antioxidant capacity, which can exacerbate BBB damage, erythrocytes may be involved in MS [73]. One important process that is specifically linked to MS is myelin formation. Researchers have looked into the potential function of mirR-219 in the myelination process using oligodendrocyte precursor cells (OPCs). Negative regulators of the myelination process were inhibited when EVs shed miR-219 into the OPC, increasing its quantity and myelin synthesis. Because miR-219 is elevated during OPC differentiation and is essential for OPC maturation and the maintenance of compact myelin, but is deficient in MS lesions, its function is representative of MS [74]. Furthermore, data showing that miR-219 was missing in the CSF of MS patients compared to healthy subjects suggests that the miRNA may be a potential biomarker of MS. This may be due to the fact that the miRNA has been sequestered inside EVs [89]. Because they are simple and noninvasive to collect, urinary EVs are another intriguing source of biomarkers. However, there are not many studies on urinary EVs in MS. In fact, only Singh’s group demonstrated the potential value of EV miRNAs obtained from urine as a biomarker. When the authors compared the EV miRNA profiles in plasma and urine during the pre-onset, onset, and peak stages of EAE disease, they discovered that EV-derived miR-155-5p was overexpressed during the pre-onset phase [75]. This miRNA modulates the autoimmune response in MS and is a potent regulator of inflammation [90]. In addition, they looked at how glatiramer acetate, which is typically used in MS therapy, affected the expression of miRNAs. They found, for the first time, that expression was modified during treatment, particularly in urinary EV miRNAs, and that miR-9-5p and miR-35-3p were significantly dysregulated at the EAE peak stage. Urinary EVs may provide molecular biomarkers of treatment response and disease progression, according to the researchers’ suggestion. Scaroni et al. identified two miRNAs packed in blood myeloid EVs as potential indicators for cognitive impairments in MS. By comparing two distinct small cohorts of MS patients with cognitive impairment versus those with cognitive preservation, they were able to measure a small number of miRNAs that may be associated with synaptic dysfunction in plasma EVs. In EVs produced from microglia, they discovered a particular profile of cognitive impairment, which consisted of low levels of let-7b-5p and high levels of miR-150-5p, while between cognitive impairment and cognitive preservation MS patients, there were no appreciable differences in these miRNAs in total plasma EVs [76].

In conclusion, the identification of miRNAs as being involved in MS should significantly enhance the illness’s diagnosis and treatment. Because miRNAs are far more persistent than mRNA or proteins, they may be a useful biomarker. In order to evaluate miRNA EVs as a potential diagnostic biomarker, future research will need to optimize and standardize procedures for isolating and characterizing the EVs.

### 5.2. EV-Derived Proteins

There is still much to learn about the EV protein cargo in MS. The most immunogenic myelin protein, myelin oligodendrocyte glycoprotein (MOG), was only expressed on the surface of myelin sheaths and the oligodendrocyte membrane. Galazka and colleagues found this protein to be strongly correlated with disease activity when they examined the EV protein content in the serum and cerebrospinal fluid of MS patients [77]. In serum-derived EVs from SPMS patients and RRMS patients during relapse, MOG was elevated. In CSF, MOG levels were higher in all MS groups without differing from controls. As a result, it seems that serum and CSF EVs have similar MOG contents.

Furthermore, the authors stimulated patient and control serum PBMCs to produce EVs in vitro in order to rule out the possibility that MOG could originate from PBMCs rather than the CNS. MOG is not present in EVs produced by PBMC cultures, despite the high volume of EVs produced. Bhargava et al.’s work provides the way for more research on the function of Toll-like receptors (TLRs) in the serum EVs of patients with RRMS. They used an immunological array to detect membrane proteins on EVs, and discovered that MS patients had lower TLR3 and TLR4 levels than controls. TLR4 appears to support an inflammatory process in MS, while TLR3 appears to play a protective role [91]. Although previous studies have indicated that these receptors have a role in regulating MS and EAE, Bhargava’s work is the first to examine the TLR concentration in EVs [78]. As was noted before, plasmatic EVs may exacerbate the inflammatory processes in MS patients [37,41]. According to Willis and colleagues, this feature might be connected to fibrinogen in the EV burden, which can cause relapses on their own in mice models of EAE. In fact, fibrinogen was identified as a significant proportion in the same work that performed a proteomics investigation on the plasmatic EV cargo of MS patients [79]. It has been determined that fibrinogen in MS is associated with inflammation and disruption of the BBB, promotes the creation of new lesions, and limits tissue regeneration [92]. Alternative biological fluids, including CSF, might provide an accessible source of protein biomarkers produced from EVs. The proteomic profile central nervous system of 442 important EV proteins that were taken out of the CSF of patients with MS and neuromyelitis optica was examined by Lee et al. This investigation demonstrated the significant presence of distinct protein signatures that can be used to distinguish between these pathologies, as well as fibronectin, which is specifically linked to MS, and glial fibrillary acidic protein, which is associated with neuromyelitis optica. Therefore, this work provides validity to the idea that the cargo of EVs may act as a biomarker resource and aid in the accurate diagnosis of neuroinflammation diseases [80]. For the first time, in the work of Pieragostino et al., proteomic analysis of EVs collected from MS patients’ tears revealed the existence of both neural- and microglia-derived EVs [81]. In addition, compared to HC, the EV protein cargo from tears was almost 70% the same as that extracted from CSF, with comparable and unique disease-specific pathways. Given that the eyes are the natural extension of the brain and that oligoclonal bands, which are used to diagnose multiple sclerosis, are found in the tears of MS patients, this evidence provides support for the possibility that tears could be a legitimate source of biologicals markers, like CSF [93,94]. Rather interestingly, oligodendrocytes are highly specialized cells that produce myelin and are the focus of inflammatory and immunological responses that are characteristic of MS [95]. Oligodendrocytes secrete EVs that can interact with other brain cells, just like other glial cell types [96,97]. Torres Iglesias et al. conducted a study to determine the EV profile produced by the immune and nervous system from blood that could have a specific role as a biomarker in MS, and identified a specific protein profile. The results suggest that these vesicles are potentially useful specific biomarkers for MS. The size of oligodendrocyte-derived EVs corresponds with motor and cognitive impairment in MS patients, while the size of T cell- and neuron-derived EVs may represent disease activity. Indeed, the oligodendrocyte, neuronal, and immune cell contents of EVs produced from blood appear to be significant specific markers for multiple sclerosis [98].

There are currently few studies on oligodendrocyte-derived EVs, most of which concentrate on their possible application as biomarkers. MBP (myelin basic protein) and MOG content in oligodendrocyte-derived EVs from serum were the topic of Agliardi et al.’s study, which proposed their use as biological markers to aid in the diagnosis of clinical MS symptoms [82].

In conclusion, while the role of EVs in MS remains incompletely understood, their potential relevance to disease pathophysiology could provide new insights into their evaluation as biomarkers in MS, treatment choice, and efficacy.

## 6. EVs for MS Therapy

As mentioned previously, specific origin markers are present on the surface of EVs, and their cargo is specifically established on both donor and target cells. They are also able to move both ways across the BBB. Because of these characteristics, EVs are attractive for use in the development of novel treatment approaches that will best restore the damaged myelin in MS patients (Table 4).

The role of EVs in the myelination process first became evident when it was shown that oligodendrocyte-derived EVs are abundant in myelin-related proteins like MOG and MBP. Subsequently, Fruhbeis et al. also demonstrated that EVs mediated the communication between oligodendrocytes and axons, which is necessary for producing myelin [97,108]. In fact, several investigations have suggested that EVs are used to aid the myelin repair process. Dendritic cell cultures release extracellular vesicles (EVs) in response to low concentrations of interferon γ (IFN γ), and the cells that preferentially absorb these EVs are oligodendrocytes, which are involved in myelination. Moreover, IFNγ-DCs-EVs administered nasally improved myelin formation in vivo [99]. Pusic et al. demonstrated that under physiological and demyelinating conditions, serum EVs activate OPCs and their differentiation to generate mature myelin. The same study showed that EVs with pro-myelinating effects were released by young and old rats subjected to environmental enrichment. Moreover, aging rats’ myelination was enhanced by the nasal delivery of serum EVs from young animals, suggesting that EVs may have applications in vivo. Of note, miRNA-219 mediated this important result by influencing oligodendrocyte differentiation and causing an increase in myelin synthesis [74]. Interestingly, the same groups recently performed immunoblots for known targets of miR-219 to further confirm that oligodendrocyte lineage cells take up nasally administered IFNγ-DC-EVs. They discovered drastically reduced expression. Together, the data reported here give more evidence to support of using nasal delivery to administer IFNγ-DC-EVs as a possible treatment to enhance remyelination [100]. Riazifar et al. conducted a study to examine the effects of intravenous administration of exosomes from mesenchymal stem cells (MSCs), stimulated by IFN-γ. The results indicated that MSC-Exo decreased neuroinflammation, increased the number of Treg cells (Regulatory T cells) in the spinal cord, and decreased demyelination and the mean clinical score of EAE mice [109].

Another method of preventing myelin degradation is the control of the immune system by currently available MS therapies, which limit autoimmune attacks on myelin sheaths. According to Zhuang et al.’s findings, EVs carrying curcumin or a Stat3 inhibitor reached the brain microglia, delaying the start of EAE. This suggests that EVs could be used as a shuttle of anti-inflammatory medications, even intranasally [101]. EVs can potentially be used as a delivery system for medications or other compounds that reduce inflammation. EAE was improved by intracisternal injection of modified microglia-derived EVs that expressed a protein that increased their absorption by myeloid cells and astrocytes and included IL-4 [102].

Chronic inflammation plays a crucial role in the pathophysiology of MS, where oligodendrocytes (remyelination) and neuronal cells (degeneration) are also involved in addition to peripheral (T and B lymphocytes) and central (microglia) immune cells. The ability of immune-cell-derived EVs to influence the immune system has drawn attention to their potential use as therapeutic agents. MSCs, which can come from different sources, possess the most interesting characteristics among stem cells regarding demyelinating diseases. Promising outcomes have been noted when MSCs are used in MS patients and animal models. When MSC-EVs are combined with a myelin-specific DNA aptamer that induces remyelination on their surface, immunomodulatory effects and remyelination are generated in EAE [110]. Due to the circulating property, EVs are a promising option for a medicinal delivery system in the future. In fact, the inflammatory responses in EAE models have been markedly decreased by in vivo injections of EVs made from MSC of various tissue types and healthy glial cells, including microglia and oligodendrocytes. The creation of modified EVs carrying specific peptide cargoes to target pathogenic cells has also been accomplished and shown to be effective in preventing the EAE model [111]. More research in this area to rely on EVs as a therapeutic agent will be fruitful for MS treatments in the future, especially in light of these encouraging data.

Axon-disruption-induced neurodegeneration is another driving mechanism of demyelinating disorders. In traumatic brain injury models, mesenchymal stem cell (MSC)-derived EVs appear to act as peripheral immunomodulators, decreasing inflammation and enhancing neuroprotection, angiogenesis, and neurological function. This suggests that EVs may one day be used therapeutically for diseases like MS, where neuroprotection needs to be reinforced [112]. Additionally, MSCs are showing promise as a therapy strategy for MS [113]. In this sense, EVs produced from MSCs are suggested to have a myelin improving function by several investigations. Clark et al. showed that by injecting EVs derived from placental MSCs into EAE animal models, EVs could promote OPC differentiation and the production of new myelin [104]. Xun and collaborators conducted a meta-analysis with the aim of systematically reviewing the efficacy of MSC-EVs in preclinical animal models of MS. A notable increase in functional outcomes in the MS animals treated with MSC-EV therapy was one of the study’s findings, which also offered insight into the possible therapeutic uses of MSC-EVs in preclinical MS research [103]. Remarkably, EVs produced from human adipose tissue-derived MSCs (hAdMSCs) administered intravenously improved motor impairment decreased brain atrophy, promoted remyelination, and controlled neuroinflammation in a primary progressive multiple sclerosis mouse model [105]. Furthermore, because hAdMSC injection stimulates natural recovery mechanisms, it has proven to be a successful therapeutic strategy for brain injury in animal models [114]. Furthermore, in the work of Jafarinia et al., intravenous administration of hAdMSC-derived EVs was shown to have therapeutic implications in EAE models, reducing T cell proliferation, leukocyte infiltration, and demyelination. However, the authors concluded that hAdMSCs had a greater effect on Treg cells (Regulatory T cells) than the corresponding EVs [106]. The work of Farinazzo et al., describing the intravenous administration of adipose stem cell-derived EVs (EV-ASCs) in an EAE animal model, seems at odds. The authors observed a protective effect of EV-ASCs only on EAE before disease onset, but not on confirmed EAE [107]. Even with this disparity, research into the potential application of MSC-derived EVs as a cell-free substitute for MSC-based therapies in the treatment of multiple sclerosis is worthwhile. Lastly, EVs’ evaluation as a diagnostic tool for disease stages is intriguing. In their research, Sáenz-Cuesta and colleagues found that EVs’ concentration increased significantly five hours after fingolimod (FGM) injection, and that their miRNA cargo changed quickly and significantly after that. Additionally, compared to EVs before to FGM therapy, the inhibitory action of free EVs on lymphocytes was lower after the first dose of FGM [67]. Given that FGM is an analog of natural sphingosine and should prevent vesicular trafficking, this was an unexpected outcome [115]. Finally, these data imply that FGM controls the release and immunological function of EVs in MS patients shortly after it is administered, so this qualifies EVs as biomarkers for early therapy monitoring.

## 7. Concluding Remarks

Over the past ten years, EVs, which are specialized structures for cellular communication, have received increasing attention. There is little doubt that the increasing number of research publications in the EVs field on neurological diseases, such as MS, is due to the following factors: (1) their ability to pass through the BBB in both directions; (2) their study in all biological fluids, indicating that they can be used in peripheral fluids like serum or plasma, that bypass the invasiveness and cost of diagnostic and prognostic tools; (3) their content, which is well enclosed inside the target cells and reflects that of the donor cells; (4) the fine recognition mechanism of target cells, which is yet unclear; (5) the presence on their surface of donor-specific molecular markers; and (6) the possibility of using EVs as biomarkers even in the early stages of the disease.

Numerous studies have looked at the possible involvement of EVs produced from several cell types in the pathophysiology of MS and in the animal model EAE. Compared to HCs, there have been reports of a higher quantity of EVs in the serum/plasma and/or CSF of MS patients. It would be highly beneficial to identify EVs as new possible predictive biomarkers, which may be used as potential therapeutic targets, since the quantity of EVs in body fluids appears to be connected with the activation of cells involved in MS pathogenesis. This is the reason that numerous studies have focused on the generation of monocyte-derived EVs, their cargo, and their pharmacological regulation.

However, their application in clinical practice is a challenge because there are still many unsolved technical problems. Pre-analytical parameters include a wide range of variables that impact EV collection and isolation. These include the types of sources that are taken into consideration (such as serum, plasma, CSF, urine, etc.), the procedures involved in processing, and storage. Currently, proteins and miRNAs are the most studied biomarkers. As dysregulated miRNAs are typically studied in clusters and their cargo varies across EVs from different origins, it is currently challenging to identify a single, distinct miRNA as an MS biomarker. Proteomic analysis appears to be a useful technique to highlight promising biomarkers, and proteins are a more stable specimen; however, there are not many studies in the literature at this time. Conversely, EVs appear to be one of the factors that also contribute to the progression of inflammation. As a result, examination of their cargo could lead to novel pathways into the pathophysiology of MS in addition to new biomarkers of disease stages and therapeutic targets.

We reviewed the available research in this review, with a particular emphasis on EVs’ function in the pathogenic mechanism underlying MS and EAE, as well as their potential application as therapeutic and diagnostic agents. EVs have been shown to be effective carriers in bidirectional communication between the peripheral and central nervous systems, actively participating in the degenerative processes of MS. EVs connect with target cells to deliver their message-cargo, which modifies the target cells’ effector activities and gene expression. In particular, their cargo material exhibits extraordinary stability and could be indicative of the parent cell’s current pathological condition. However, standardization of EV extraction and purification techniques, as well as more precise and efficient quantification of EV concentration or quantity, remain issues to be resolved. EVs characterization is a challenge for researchers, and the approaches that are currently employed are essentially imprecise.

The development of new technologies and standardized protocols for EV isolation and characterization will permit a comprehensive knowledge of EVs’ role in the pathophysiological processes associated with MS. More work needs to be carried out to ascertain the characteristics of their origin and the pathogenicity role of EVs, but new technologies that can address this issue more effectively may open up the gate to new diagnostic techniques.

## Figures and Tables

**Table 1 genes-15-00772-t001:** Most widely used techniques for isolating EVs, with advantages and disadvantages.

Isolation Technique	Advantages	Disadvantages
Differential centrifugation	This method offers a high yield, is low-cost, and is simple to use.	Produces excessive contaminants, necessitates specialized equipment, and may destroy EVs.
DGU	DGU generates very pure EVs at low cost.	Requires specialized tools and training, and it has low yield, low scalability, and low throughput.
SEC	Scalable efficient in achieving excellent separation, maintaining EV integrity, and eliminating soluble proteins and small compounds.	Sample dilution is necessary for SEC due to its low sample capacity.
Bidimensional:DGU + SEC	These methods yield extremely high purity.	Has a low yield, limited throughput, and little scalability and requires specialized equipment and training.
Precipitation	This method is quick, simple to use, fast, scalable, and economic.	Low purity is produced by precipitation, soluble non-EV material is isolated, and the precipitation reagent must be eliminated.
Immunoisolation	This method generates extremely pure EVs quickly and does not need specialized equipment.	Immunoisolation.

Abbreviations: DGU, density gradient ultracentrifugation; SEC, size exclusion chromatography.

**Table 2 genes-15-00772-t002:** Characteristics of EVs and potential role in MS.

EV Origin	Markers	Source	Technique	Potential Application	Reference
Endothelial cells	CD31, CD51	PPP/MVEC	FC	Endothelial dysfunction	[37]
Endothelial cells	CD31, CD62	PPP/MVEC	FC	Endothelial dysfunction	[38]
Endothelial cells	CD4, CD8	MVEC	FC	T-lymphocytes activation	[39]
Platelet	CD31, CD42b	PPP	FC	BBB dysfunction	[40]
	CD54, CD62e	WB/MVEC	FC	Disease activity	[41]
Platelet	CD3, CD41,	PFP	SEC/WestB	Disease activity	[42]

BBB, blood–brain barrier; PPP, platelet poor plasma; PFP, platelet free plasma; WB, whole blood; SEC, size exclusion chromatography; WestB, Western blot; FC, flow cytometry; MVEC, microvascular endothelial cell culture.

**Table 3 genes-15-00772-t003:** Molecular EVs’ cargo detected in MS patients, and in vivo and in vitro models.

EVs Biological Source	EV Cargo	References
	**miRNAs**	
Serum	miR-15b-5p, miR-23a-3p, miR-223-3p, miR-374a-5p, miR-30b-5p, miR-433-3p, miR-485-3p, mir-342-3p, miR-432-5p	[68]
Serum	miR-122-5p, miR-196b-5p, miR-301a-3p, miR-532-5p, miR-23a, miR-15b, miR-223	[66]
Serum	miR-146a, mir-451, miR-26a, let7	[69]
T cell cultures	miR-326	[70]
Plasma	miR-25, miR-19b, miR-29a, let7i	[71]
Activated microglia	miR-146a-5p	[72]
Erythrocyte	miR-451a	[73]
OPC cells	miR-219	[74]
Urine of EAE models	miR-155-5p, miR-9-5p, miR-35-3p	[75]
Myeloid cells	miR-150-5p and let-7b-5p in cognitively impaired MS patients vs. cognitively preserved MS patients	[76]
	**Proteins**	
Serum and CSF	MOG	[77]
Serum	TLR3 TLR4	[78]
Plasma of EAE models	Fibrinogen	[79]
CSF	Fibronectin GFAP, Integrin signaling events, PI3K signaling, EGF receptor (ErbB1) signaling pathway, ErbB receptor signaling network, IFN pathway, LKB1 signaling events, PDGF.	[80]
Tears and CSF	receptor signaling network, RNA polymerase I, S1P1 pathway, signaling events mediated by VEGFR1 and VEGFR2, TNFα/NFkB, TRAIL signaling pathway.	[81]
Serum	MBP levels in EVs in CIS, RRMS, PPMS vs. HC MBP levels in EVs in PPMS vs. RRMS and CIS.	[82]

EVs: extracellular vesicles; miR-: microRNA; OPC: oligodendrocyte precursor cell; EAE: experimental autoimmune encephalomyelitis; CSF: cerebrospinal fluid; MOG: myelin oligodendrocyte glycoprotein; TLR3 and TLR4: toll-like receptors 3 and 4; GFAP: glial fibrillary acidic protein; EGF: epidermal growth factor; S1P: sphingosine 1phosphate; IFN: interferon; LKB1: liver kinase B1; PDGF: platelet-derived growth factor; VEGFR1: vascular endothelial growth factor receptor1; VEGFR2: vascular endothelial growth factor receptor1; TRAIL: TNF-related apoptosis-inducing ligand; TNFα/NFkB: tumor necrosis factor-α/nuclear factor-κB; MBP: myelin basic protein; CIS: clinically isolated syndrome; RRMS: relapsing remitting multiple sclerosis; PPMS: primary progressive multiple sclerosis.

**Table 4 genes-15-00772-t004:** Therapeutic applications of EVs in MS.

Source of EVs	Effect	References
EVs from DCs	Nasal administration of IFNγ-DCs-EVs enhanced myelin generation in vivo.	[99]
EVs from young EE rats	Nasal administration of serum EVs from young animals increased myelination in aging rats.	[74,100]
EVs from glioblastoma cell line (GL26)	Nasal administration of EVs containing curcumin or Stat3 inhibitor reached brain microglia, delaying EAE.	[101]
microglia BV-2	IL-4+ EVs induced, in vitro, the upregulation of anti-inflammatory markers in different myeloid cells.	[102]
MSC-EVs from EAE	Animals benefited from therapy with MSC-EVs, demonstrating a significant improvement in functional outcomes.	[103]
EVs from pMSCs	Injection in EAE models of EV pMSCs stimulated OPC differentiation, promoting new myelin formation.	[104]
EVs from hAdMSCs	Intravenous administration of EVs from hAdMSCs in a primary progressive MS murine model ameliorated motor disability, reduced brain atrophy, and promoted remyelination, regulating neuroinflammation.	[105]
EVs from hAdMSCs	Intravenous administration of EVs from hAdMSCs reduced T cell proliferation, leukocyte infiltration, and demyelination.	[106]
EVs from ASCs	Intravenous administration of EV-ASCs in EAE model had a protective effect before EAE onset, but not on established EAE.	[107]

EVs: extracellular vesicles; DC: dendritic cells; EE: environmental enrichment; hAdMSCs: human adipose mesenchymal stem cells; ASCs: adipose stem cells; EAE: experimental autoimmune encephalomyelitis; pMSCs: primary mesenchyme cells; OPC: oligodendrocyte progenitor cell.

## Data Availability

No new data were created or analyzed in this study. Data sharing is not applicable to this article.

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
