# Peer review of "Extracellular Vesicles in Multiple Sclerosis: Their Significance in the Development and Possible Applications as Therapeutic Agents and Biomarkers"

_genes, 2024, doi:10.3390/genes15060772_

Round 1

Reviewer 1 Report

Comments and Suggestions for Authors

In this review Manna et all summarise evidences supporting Extracellular Vesicles (EV) involvement in the pathophysiology of Multiple Sclerosis. 

The review touches several aspects of the biology of EV in relation to both the pathogenesis of MS, including their involvement in both inflammatory and neuroprotective pathways, their potential use as diagnostic or prognostic biomarkers, and their potential therapeutic use.

The topic is complex, and the review is comprehensive, however it would benefit from major adjustments. Data are described superficially. Sentences like “a role has been described” or “according to reports” or “play a role” and so on should be avoided, to ensure concise presentation of scientific data. 

Data should be presented in an hypothesis based structure. At the moment there are four main paragraphs: 

- dual role of EV in MS

- EV in MS pathogensesis

- EV as biomarkers

- EV for MS therapy

From none of them is clear what the message / conclusions are and at the moment each paragraph contains a huge set of unorganised data. An improvement would already come from the division of each paragraph into subparagraphs where the authors try to answer specific questions about the EV. The simple expression “EV are involved in MS” is not enough. 

The paragraph about MS contains imprecise if not wrong information, the authors are kindly asked to improve it with more appropriate terminology. 

A couple of figures would provide a strong benefit to this review manuscript. 

Comments on the Quality of English Language

No major corrections of english required, but clearly written from non-english speaker.

Author Response

Response to Reviewer.

Comments to Reviewer #1:

Thank you very much for the interest in our manuscript and for all your suggestions.

We sincerely thank the reviewer for this comment. Following her/his suggestion, a revision has been made, so we hope that this version of the manuscript resulted more fluent to read.

In response to your specific suggestions: we have made all the corrections suggested.

Questions: The topic is complex, and the review is comprehensive, however it would benefit from major adjustments. Data are described superficially. Sentences like “a role has been described” or “according to reports” or “play a role” and so on should be avoided, to ensure concise presentation of scientific data.

Answer: We thanks for the suggestion, we avoided these sentences.

Question: Data should be presented in an hypothesis based structure. At the moment there are four main paragraphs ……………………From none of them is clear what the message / conclusions are and at the moment each paragraph contains a huge set of unorganised data. An improvement would already come from the division of each paragraph into subparagraphs where the authors try to answer specific questions about the EV. The simple expression “EV are involved in MS” is not enough

Answer: We have done a review and implementation of each paragraph, so we hope that this version of the manuscript will be more organised and smooth to read. We have divided the paragraph “EVs as biomarkers for MS” into subparagraphs.

Question: The paragraph about MS contains imprecise if not wrong information, the authors are kindly asked to improve it with more appropriate terminology.

Answer: We have provided more appropriate terminology regarding what you suggested.

Question: A couple of figures would provide a strong benefit to this review manuscript

Answer: We decided not to include figures because we don't think they are necessary for this review.

Question: No major corrections of english required, but clearly written from non-english speaker

Answer: An English lecturer revised the English grammar of the entire manuscript, so we hope that this version is easier to read.

Reviewer 2 Report

Comments and Suggestions for Authors

Manna et al provide a review on the roles of extracellular vesicles (EVs) in multiple sclerosis. This is an emerging and important topic. However, there is still much room for improvement with regard to this manuscript.

1. A large part of the first section and the Table 1 and Table 2 are too general to be included in this article. I would suggest that the authors remove these materials and refer the readers to a few representative methodology papers instead.   

2. In the discussion of EVs as therapeutic agents for MS (section 2.4.), it is interesting to note that different EVs have different cellular targets. For example, some target oligodendrocytes (IFNγ-DC-EVs, page 12, line 447) and some target microglia (page 12, line 453). It would be better if the authors discuss in more detail about the mechanisms and utilities of different targeting properties among various potential EVs therapeutics.   

3. Several prominent and relevant papers are not included in this review (some examples are given below). It appears to me that the literature search was not up to date.   

1) https://pubmed.ncbi.nlm.nih.gov/32669030/ Synaptic and complement markers in extracellular vesicles in multiple sclerosis. Mult Scler. 2021 Apr;27(4):509-518. doi: 10.1177/1352458520924590. Epub 2020 Jun 17.  

2) Brain and immune system-derived extracellular vesicles mediate regulation of complement system, extracellular matrix remodeling, brain repair and antigen tolerance in Multiple sclerosis. Brain Behav Immun. 2023 Oct:113:44-55. doi: 10.1016/j.bbi.2023.06.025. Epub 2023 Jul 3.  

3) Mitochondrial measures in neuronally enriched extracellular vesicles predict brain and retinal atrophy in multiple sclerosis. Mult Scler. 2022 Nov;28(13):2020-2026. doi: 10.1177/13524585221106290. Epub 2022 Jul 5.  

4) CNS endothelial derived extracellular vesicles are biomarkers of active disease in multiple sclerosis Fluids Barriers CNS. 2022 Feb 8;19(1):13. doi: 10.1186/s12987-021-00299-4.

4. abstract (line 20-21): "EVs can be ...... therapy targets". My opinion is that EVs could be potential therapeutic agents, but their roles as therapeutic targets in MS are, in any case, immature at present.   

5. abstract (line 11): intracellular? intercellular?  

6. Some citations are redundant, such as ref.108 and 109.  

7. "Supplementary Materials: The following supporting information can be downloaded at:  https://www.mdpi.com/xxx/s1, Figure S1: title; Table S1: title; Video S1: title." -> This part should be removed. 

Comments on the Quality of English Language

Moderate editing of English language required

Author Response

Comments to Reviewer #2:

Thank you very much for the interest in our manuscript and for all your suggestions.

We sincerely thank the reviewer for this comment. Following her/his suggestion, a revision has been made, so we hope that this version of the manuscript resulted more fluent to read.

In response to your specific suggestions: we have made all the corrections suggested.

Question: A large part of the first section and the Table 1 and Table 2 are too general to be included in this article. I would suggest that the authors remove these materials and refer the readers to a few representative methodology papers instead

Answer: In this version of the manuscript we have eliminated table 1 and table 2, as suggested.

Questions: In the discussion of EVs as therapeutic agents for MS (section 2.4.), it is interesting to note that different EVs have different cellular targets. For example, some target oligodendrocytes (IFNγ-DC-EVs, page 12, line 447) and some target microglia (page 12, line 453). It would be better if the authors discuss in more detail about the mechanisms and utilities of different targeting properties among various potential EVs therapeutics.  

Answer: We have added some information about EVs as therapeutic agents, and restructured all of the paragraphs in this version of the manuscript.

Question : Several prominent and relevant papers are not included in this review (some examples are doi: 10.1177/1352458520924590., doi: 10.1016/j.bbi.2023.06.025, doi: 10.1177/13524585221106290, doi: 10.1186/s12987-021-00299-4). It appears to me that the literature search was not up to date.  

Answer: We regret that some relevant papers were missing, and this has been addressed in this version of the manuscript. We have included all the suggested references.

Question : abstract (line 20-21): "EVs can be ...... therapy targets". My opinion is that EVs could be potential therapeutic agents, but their roles as therapeutic targets in MS are, in any case, immature at present.  

Answer: We agree whit you, we thanks for the suggestion. We have explained the above statement better.

Questions : abstract (line 11): intracellular? intercellular? 

Answer: We apologize for this mistake, and in this version of the manuscript, we have corrected it, according to your suggestion.

 Questions : Some citations are redundant, such as ref.108 and 109.

 Answer: We apologize for the mistake, and we have carefully looked at all references.

Questions : "Supplementary Materials: The following supporting information can be downloaded at:  https://www.mdpi.com/xxx/s1, Figure S1: title; Table S1: title; Video S1: title." This part should be removed.

 Answer: We apologize for this mistake, however we never added the Supplementary Materials.

 Questions : Moderate editing of English language required

 Answer: An English lecturer revised the English grammar of the entire manuscript, so we hope that this version is easier to read.

Round 2

Reviewer 2 Report

Comments and Suggestions for Authors Although the authors added the findings of several recent papers into the revised manuscript, the review remains loosely organized. Besides, many mistakes could be seen with regard to the grammar and wording and spelling.   1. I would suggest that the authors prepare a Table to systematically examine existing EV research in patients with MS. Specifically, the cellular origins (neuron / astrocyte / oligodendrocyte / microglia / endothelial cell / various immune cells, and their associated markers), cellular targets (and associated markers), cargos, basic and clinical implications (such as complement involvement, or specific EVs/biomarkers as surrogate for disease activity / cognitive impairment ....), and other relevant information (such as methods of EV isolation/detection) could be presented in such a table, which allows for easier comparisons.    2. I would suggest that the authors critically appraise the existing literature, not only highlighting important findings but also identifying their insufficiency and knowledge gaps. Comments on the Quality of English Language

Please check grammar and wording throughout - mistakes are frequently seen.

Author Response

Response to Reviewer.

Comments to Reviewer #2:

Thank you very much for the interest in our manuscript and for all your suggestions.

We sincerely thank the reviewer for this comment. Following her/his suggestion, a revision has been made, so we hope that this version of the manuscript resulted more fluent to read.

In response to your specific suggestions:

  • We hope that this version of the manuscript will be easier to read  and more well organized, since as each paragraph has been implemented.
  • We have provided more appropriate terminology regarding what you suggested.
  • In this version of the manuscript we have added, as suggested, a table reporting the most common methods of isolation/detection of EVs and a table related to their cellular origin, cellular targets, etc.
  • We apologize for grammatical and spelling errors. We have carefully reviewed the entire manuscript.